# Exploring the potential impact of medical errors research on population health

Mabel Adelvia Sarquis Rivera[1], David A. Hernandez-Paez[2], Johana Galván-Barrios[3], Ernesto Barceló-Martinez[4], Alexis Rafael Narvaez-Rojas[5,6]*, Ivan David Lozada-Martinez[3,7]*

1 Universidad de la Costa, Barranquilla, Colombia, 2 Center for Meta-Research and Scientometrics in Biomedical Sciences, Barranquilla, Colombia, 3 Biomedical Scientometrics and Evidence-Based Research Unit, Department of Health Sciences, Universidad de la Costa, Barranquilla, Colombia, 4 Clínica Colsanitas S.A., Clínica El Carmen, Barranquilla, Colombia, 5 Hospital Militar Escuela "Alejandro Davila Bolaños", Managua, Nicaragua, 6 Maimonides Cancer Center, Brooklyn, New York, United States of America, 7 Clínica Colsanitas S.A., Clínica Iberoamérica, Barranquilla, Colombia

* ilozada@cuc.edu.co (IDL-M); axnarvaez@gmail.com (ARN-R)

## Abstract

### Background

While most research on medical errors has focused on reducing these events within clinical settings, little is known about whether this scientific research translates into improvements in population-level health or system indicators. This study aimed to explore the potential impact of medical errors research on population health, health system, and research and development indicators.

### Methods

A longitudinal analysis was conducted using global data from 1995 to 2024. Annual publication counts on medical errors were matched with 18 global population and structural indicators across four domains: mortality, health systems, research and development, and financial risk. Countries were stratified into income groups, and associations were analysed using fixed-effects, negative binomial, and hierarchical mixed-effects models.

### Results

Higher research output on medical errors was associated with reductions in neonatal, infant, under-5, and adult mortality, particularly in high-income countries and upper-middle-income countries (UMICs). Significant associations were also found with reduced risk of catastrophic and impoverishing surgical expenditures in UMICs and low- and middle-income countries. Modest links were observed with hospital bed density and intellectual property flows. However, no consistent associations were

**Data availability statement:** All relevant data are within the manuscript and its Supporting information files.

**Funding:** The author(s) received no specific funding for this work.

**Competing interests:** The authors have declared that no competing interests exist.

found in low-income countries or in hierarchical models adjusting for income-level heterogeneity.

## Conclusions and implications

Scientific research on medical errors shows potential to influence key population health- and structural-level indicators, particularly in countries with developing research ecosystems. These findings address a critical knowledge gap by providing quantitative evidence of research relevance beyond academic metrics. Promoting equitable research capacity and translation may enhance the real-world impact of patient safety efforts globally.

## Introduction

Over the past two decades, the growing global focus on improving patient safety has brought medical errors to the forefront of health policy discussions [1]. These events, often preventable, continue to affect millions of lives, undermine public trust, and impose a heavy burden on health systems worldwide [2,3]. Despite increasing visibility in clinical and policy scenarios [3], medical errors remain a complex and under-addressed phenomenon, especially when considering their broader impact beyond the individual level [4].

While much of the literature has focused on the identification, classification, and reduction of medical errors within institutions [5], relatively little is known about the potential influence of research on medical errors over population-level indicators. In other words, can the scientific effort to understand and reduce medical errors shape measurable outcomes in population health, health systems performance, or innovation? And if so, is this influence uniform across countries with different economic realities?

Previous systematic reviews have shown that certain interventions can reduce the incidence and financial burden of medical errors on healthcare systems, typically measured through local healthcare quality indicators [6]. It would be reasonable to expect this trend to hold when assessing macro-level indicators. However, this relationship has not yet been quantified. As a result, a theoretical, empirical and practical-knowledge gaps persist, hindering the development of recommendations grounded in robust quantitative findings [7].

Historically, the impact of health research has been assessed using metrics such as citations, policy uptake, or shifts in clinical guidelines [8]. Yet, few studies have attempted to bridge the gap between knowledge production in medical error research and real-world indicators like mortality, health spending, financial protection, or research capacity [9,10]. This disconnect poses a key question: to what extent does scientific inquiry into medical errors translate into tangible improvements in health systems or population health?

Although a substantial body of literature has evaluated medical errors and patient safety at the clinical or institutional level, no prior study has systematically examined

whether scientific research on medical errors translates into changes in national-level health, financial, and innovation indicators across countries and over time. In particular, the lack of longitudinal, cross-country, and multi-domain analyses represents a critical gap in understanding the real-world impact of patient safety research beyond academic or hospital-based metrics.

This question is particularly pressing in low- and middle-income countries (LMICs), where limited research output may reflect broader systemic barriers, and where the consequences of inaction can be most severe [11]. On the other hand, in high-income countries (HICs) with greater resources, the assumption that research leads to impact remains largely untested across macro-level indicators.

In this context, this study aims to explore whether and how scientific research on medical errors is associated with global indicators of population health, health system performance, and research investment. Using a longitudinal, income-stratified approach, this work seeks to identify potential disparities in impact across income groups, and to assess whether the global scientific community's efforts in this area are contributing meaningfully to health improvements.

By linking bibliometric data with health and development indicators across countries and time, this study addresses a significant gap in the literature: the lack of empirical evidence on the population-level consequences of medical error research [12]. In doing so, it contributes to the broader dialogue on the relevance, accountability, and equity of health research in global contexts [13].

## Methods

### Study design

Retrospective longitudinal study. Through this design, a temporal perspective was applied to explore potential associations between research activity and global population indicators [14].

### Data sources

Scientific publications related to medical errors were identified through a structured search in five major databases: PubMed/MEDLINE, Scopus, Web of Science Core Collection, SciELO Citation Index, and KCI-Korean Journal Database. These databases were selected due to their broad international coverage and their inclusion of peer-reviewed journals in the health and biomedical sciences. The use of these resources has been previously validated and reproduced in studies of this nature [9,15].

In parallel, additional types of indicators related to healthcare quality indicators and global health metrics were collected. Quantitative variables directly associated with health expenditures, disease burden, and research and development activities were analyzed. Health and development data were extracted from the World Bank Open Data platform [16] using the wbstats package in R software, while research-related indicators were obtained from the Global Observatory on Health Research and Development (R&D) [17], hosted by the World Health Organization (WHO).

All data were retrieved in July 2024. For each publication, bibliographic metadata such as title, year, authors, country of affiliation, and journal were downloaded in CSV format. Similarly, country-level indicators were downloaded in structured formats and organized by year. Country names were standardized and each one was assigned to a World Bank income group: Low-income countries (LICs), LMICs, upper-middle-income countries (UMICs), or HICs, based on the 2024 classification [18].

To ensure consistency across the 1995–2024 period, both bibliometric and indicator data were obtained from internationally curated sources that apply harmonized definitions and retrospective standardization. Bibliometric records were retrieved using reproducible search strategies across five major databases with stable indexing practices, while all national indicators were extracted from the World Bank and the WHO's Global Observatory on Health Research and Development, which provide longitudinally comparable data. All datasets were retrieved within a single time window (July 2024), and income classifications were standardized using the 2024 World Bank grouping to avoid temporal reclassification bias.

This dual data collection process made it possible to link research activity on medical errors with national-level outcomes in health, financing, and innovation, across countries and over time.

## Search strategy

A structured search strategy was designed to identify scientific publications focused on the analysis, discussion, or evaluation of medical errors. Controlled vocabulary terms (MeSH) and equivalent keywords were used to develop the search syntax. The strategy was adapted to the format of each database and focused on literature published in the health sciences, including medicine, nursing, dentistry, and related biomedical fields.

In this study, medical errors research was defined as scientific literature whose primary objective was to analyze, quantify, classify, or interpret errors occurring in the delivery of health care, including diagnostic errors, medication errors, surgical and procedural errors, and system-level or organizational patient safety failures. This definition guided both the construction of the search strategy and the eligibility assessment of retrieved records.

An initial round of pilot searches was conducted to refine the combination of terms, ensuring optimal sensitivity and precision. The final search strategy was implemented on July 15, 2024, and applied in both English and Spanish. Publications of any year were considered eligible, as no restrictions were applied to the publication date.

To be included, documents had to meet the following criteria: A) Scientific articles published in peer-reviewed journals; B) Full-text availability; and C) A clearly defined objective related to the analysis, interpretation, or evaluation of medical errors in health care.

A "clearly defined objective related to medical errors" was operationalized as a publication in which the primary stated aim, as reported in the title, abstract, or objectives section, was to analyze, quantify, classify, interpret, or evaluate medical errors, patient safety events, or error-related processes in health care. Articles in which medical errors were only mentioned incidentally or as a secondary context were excluded. Full-text availability was required not for data extraction, but to allow verification of this eligibility criterion and to prevent the inclusion of false-positive records generated by keyword-based searches alone.

Additionally, documents were excluded if they met at least one of the following conditions: A) Conference abstracts, book chapters, books, errata, or retracted articles; B) Missing essential bibliographic information (e.g., author, journal, or publication year); and C) Articles in press at the time of retrieval.

Articles written in languages other than English or Spanish were included if they contained an abstract in either of these two languages and met all inclusion criteria.

The screening process was conducted in two phases. In the first phase, titles and abstracts were reviewed independently by two researchers to verify eligibility (July 20, 2024). In the second phase, a more detailed assessment of full-text availability, metadata completeness, and document relevance was performed (September 26, 2024, to November 15, 2024). Any disagreements were resolved by a third reviewer.

An example of the final strategy, which was implemented in the Scopus database and provided the most precise results, is as follows: SUBJAREA(HEAL) OR SUBJAREA(DENT) OR SUBJAREA(NURS) OR SUBJAREA(MEDI) OR SUBJAREA(BIOC) OR SUBJAREA(IMMU) OR SUBJAREA(NEUR) OR SUBJAREA(PHAR) AND TITLE("Medical Errors") OR TITLE("Diagnostic Errors") OR TITLE("Medication Errors") This strategy was adapted for use in each of the other databases or search engines (S1 File).

A total of 2639 publications were selected for analysis. The flow of document selection is shown in **Fig 1**.

## Data standardization

To ensure consistency and quality of the dataset, a two-step review process was applied. First, duplicate records were removed, and titles and abstracts were manually screened to confirm that inclusion and exclusion criteria had been met. This screening was performed independently by two reviewers using Microsoft Excel 2016. Second, a more detailed

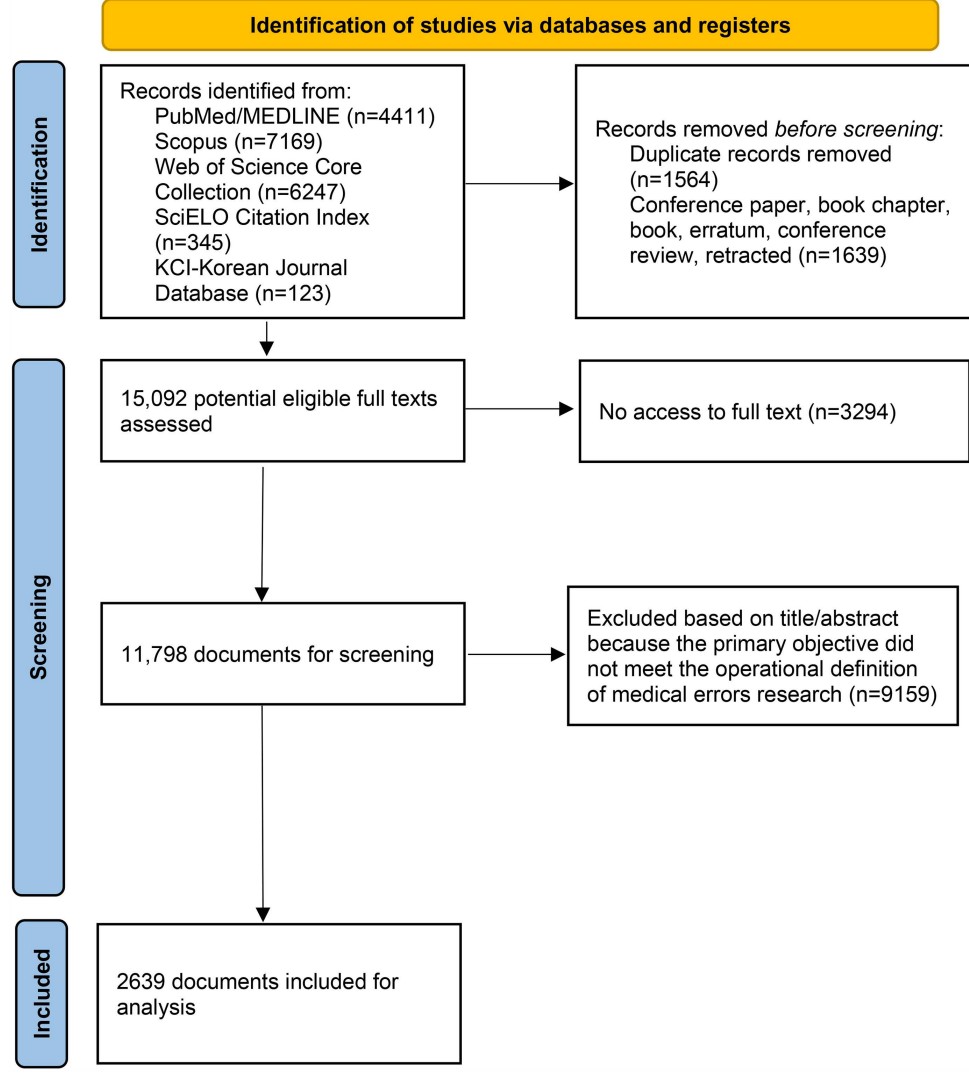

**Fig 1. Flowchart of selected documents.**

verification was conducted to complete missing values, identify inconsistencies, and harmonize key variables across data sources.

Any discrepancies between reviewers were resolved by a third researcher. Review articles, regardless of whether they followed a narrative or systematic approach (with or without meta-analysis), were grouped into a unified "reviews" category. Likewise, each publication was assigned to a country according to the institutional affiliation of the corresponding author, and the income classification of that country was standardized based on the 2024 World Bank list [18].

In parallel, national-level indicators retrieved from the World Bank [17] and WHO Global Health Observatory [19] were cleaned and reorganized for integration. Each indicator was labeled with a unique identifier, and its values were arranged by country and publication year. For alignment with the bibliometric dataset, income group names were standardized to allow for accurate merging of the datasets.

A total of 22 national indicators were considered for analysis. Indicators with more than 25% missing data were excluded (n = 6), resulting in a final set of 16 indicators. These were grouped into four thematic domains: Health System (4 indicators), Mortality (5 indicators), Financial Risk (2 indicators), and Research and Innovation (5 indicators).

The final set of 16 indicators was derived from a predefined conceptual framework comprising four complementary domains through which medical error research could plausibly influence society: population health (mortality), health system capacity, financial risk protection, and research and innovation. Within each domain, all relevant indicators available from the World Bank and WHO were initially considered. Indicators with more than 25% missing data were excluded to ensure longitudinal stability and statistical validity, resulting in a set of indicators that was both theoretically representative and empirically robust.

The full list of indicators and missing value proportions is provided in **Table 1**.

## Data synthesis

For each publication, quality metrics were collected, including the journal's quartile ranking and h-index, adjusted to the year of publication. These metrics were obtained from two platforms: Scimago Journal & Country Rank (available since 1999) and Journal Citation Reports (available since 1997). When both sources were available, the highest-ranking metric was selected.

Each document was assigned to a country based on the institutional affiliation of the corresponding author. Countries were then categorized into four income levels (HICs, UMICs, LMICs, and LICs).

To ensure alignment between bibliometric data, and health and population indicators, country-level variables were grouped by income level and publication year. For each year and income group, mean values were calculated across all available indicators (S2 File). This allowed the datasets to be joined using income group and year as common keys, after standardizing nomenclature across sources.

A left join operation was used to merge the datasets, preserving all bibliometric records and integrating only those indicators with complete or near-complete data. The resulting matrix enabled the analysis of associations between publication activity and trends in national-level health and population indicators.

After merging bibliometric and indicator datasets using a left join, remaining missing values were handled using an available-case (variable-specific denominator) approach. Each regression model was fitted using all observations available for the specific indicator, income group, and year involved in that model, without imputation or global complete-case deletion.

This structure also allowed comparisons across income groups, supporting the evaluation of possible differences in scientific impact depending on economic context.

## Statistical analysis

All statistical analyses were performed using R software (version 4.5.0) [20]. The distribution of continuous variables was assessed using the Kolmogorov–Smirnov test to determine whether they followed a normal distribution.

According to their distribution, continuous variables were summarized using mean and standard deviation (for normally distributed data) or median and interquartile range (for skewed data). Categorical variables were reported using absolute counts and percentages.

## Regression model selection

We employed an adaptive modeling approach [21] to assess relationships between publications volume and indicators. For each dependent-independent variable pair, we systematically determined the most appropriate regression model based on the underlying data characteristics. For count data, variance-to-mean ratios were used as an initial screening heuristic; however, final model selection between Poisson and negative binomial regression was based on model-based diagnostics, including residual deviance, dispersion statistics, and Akaike Information Criterion.

**Table 1. Proportion of missing values within each indicator (1995–2024).**

| Indicator* | Regression Model | MVs (%) |
|---|---|---|
| Health researchers (in full-time equivalent), as a proportion of all researchers | Excluded (>25% missing data) | 72.1 |
| Number of grants for biomedical research by funder, type of grant, duration and recipients (World RePORT) | Excluded (>25% missing data) | 67 |
| Official development assistance for medical research and basic health sectors per capita, by recipient country | Excluded (>25% missing data) | 64.5 |
| Cause of death, by non-communicable diseases (% of total) | Excluded (>25% missing data) | 86 |
| Cause of death, by communicable diseases and maternal, prenatal and nutrition conditions (% of total) | Excluded (>25% missing data) | 86 |
| Current health expenditure (% of GDP) | Beta | 12.6 |
| Death rate, crude (per 1,000 people) | Linear | 5 |
| Hospital beds (per 1,000 people) | Linear | 13.9 |
| Mortality rate, neonatal (per 1,000 live births) | Linear | 5 |
| Mortality rate, under-5 (per 1,000 live births) | Linear | 5 |
| Mortality rate, infant (per 1,000 live births) | Linear | 5 |
| Mortality rate, adult, female (per 1,000 female adults) | Linear | 5 |
| Mortality rate, adult, male (per 1,000 male adults) | Linear | 5 |
| Out-of-pocket expenditure per capita (current US$) | Linear | 12.6 |
| Physicians (per 1,000 people) | Negative Binomial | 10.1 |
| Risk of catastrophic expenditure for surgical care (% of people at risk) | Beta | 21.5 |
| Risk of impoverishing expenditure for surgical care (% of people at risk) | Beta | 21.5 |
| Specialist surgical workforce (per 100,000 population) | Excluded (>25% missing data) | 64.5 |
| Charges for the use of intellectual property, payments (BoP, current US$) | Linear | 1.2 |
| Charges for the use of intellectual property, receipts (BoP, current US$) | Linear | 1.2 |
| Research and development expenditure (% of GDP) | Negative Binomial | 6.3 |
| Researchers in R&D (per million people) | Negative Binomial | 6.3 |

BoP: Balance of Payments; GDP: Gross Domestic Product; MV: Missing Values; R&D: Research and Development.

* Indicators were extracted from the open-access databases of the World Bank, the World Health Organization's Global Health Observatory, and the World Health Organization's Global Observatory on Health Research and Development.

Binary outcomes were modeled using logistic regression with a binomial error distribution and logit link function, while continuous proportion data constrained between 0 and 1 were analyzed using beta regression with a continuity-corrected transformation to map proportions into the open interval (0,1). Specifically, observed proportions were adjusted using a standard sample-size-based transformation that shifts 0 and 1 values slightly inward before model fitting. For continuous variables with approximately normal distribution, we employed standard linear regression models. Full results can be found in S3 File.

Because income-stratified models may be unstable in settings with small sample sizes, particularly in low-income groups, we complemented these analyses with hierarchical mixed-effects models that pool information across income groups while accounting for between-group heterogeneity. This approach reduces the risk of separation and small-sample bias while preserving income-level structure.

## Income-stratified regression analyses

All regression analyses were stratified by World Bank income classification to account for the heterogeneity in health systems and research infrastructure across economic contexts. For each income group, we constructed separate models with a minimum threshold of five observations per model to ensure adequate statistical power.

We conducted two primary sets of analyses: 1) regression models with some indicators as dependent variables and publication counts as the independent variable, and 2) regression models with publication counts as the dependent variable and some other indicators as independent variables.

The bidirectional modeling strategy was designed to address two complementary meta-research questions. First, we examined whether scientific production on medical errors was associated with changes in population health, health system, financial, and innovation indicators (research impact). Second, we assessed whether health system and R&D characteristics were associated with the volume of medical error publications produced (determinants of research production). These two directions represent analytically distinct processes rather than inverse specifications of a single causal model.

## Statistical inference and multiple comparisons

For each regression model, we calculated appropriate fit statistics, including coefficients, standard errors, test statistics (t or z values depending on model type), and p-values. For linear models, we reported R-squared and adjusted R-squared values to assess explained variance.

For generalized linear models, we calculated pseudo-R-squared values (1 − residual deviance/null deviance) as appropriate measures of model fit, alongside Akaike Information Criterion (AIC) values [22].

To address multiple testing concerns, p-values were adjusted using the Benjamini-Hochberg method [23] to control the false discovery rate. Results were considered statistically significant at an adjusted p-value threshold of 0.05.

## Hierarchical mixed-effects modeling

To account for the clustered nature of our data within income groups while examining relationships between research output and health indicators, we implemented a hierarchical mixed-effects modeling approach [24]. This analytical framework allowed us to estimate both fixed effects of predictors and random effects associated with income group membership, thereby addressing potential correlation structures within economic classifications.

We constructed two primary sets of models: 1) Some indicators as dependent variables with publication counts as the independent variable, and 2) publication counts as dependent variable with some other indicators as independent variables.

For each dependent-independent variable pair, we implemented a systematic two-step modeling procedure. First, we fitted a simpler mixed-effects model with country income level (HIC, UMC, LMC, LIC) specified as a random effect, allowing for baseline differences between income groups while estimating the overall effect of the independent variable.

Second, we constructed an expanded model that incorporated interactions between the independent variable and income level to test for differential relationships across economic contexts. Model selection between simple and interaction specifications was based on AIC, with the lower-AIC model retained for inference. We required a minimum of 20 observations per model to ensure adequate statistical power.

The appropriate distributional family for each model was systematically determined based on the characteristics of the dependent variable. For continuous outcomes, we employed linear mixed-effects models (LMM) with Gaussian error structure using the lmer function from the lme4 package [25]. For count data, we implemented generalized linear mixed models (GLMM) with Poisson distribution; when overdispersion was detected (variance substantially exceeding the mean), we specified negative binomial distribution using the glmmTMB package [26].

 

For proportional data constrained between 0 and 1, we initially attempted beta regression with appropriate transformation to ensure exclusive (0,1) range; however, for indicators exhibiting convergence issues under beta distribution, we implemented logit transformation followed by Gaussian modeling.

All models included publication year as a covariate to account for temporal trends. To address multiple testing concerns, p-values were adjusted using the Benjamini-Hochberg false discovery rate method. Model diagnostics included assessment of convergence, examination of residual plots for homoscedasticity, and identification of influential observations. For models exhibiting convergence or estimation issues (particularly those with non-positive definite Hessian matrices or NaN standard errors), we implemented alternative specifications, including distribution changes, interaction term removal, or transformation of the dependent variable. Full results can be found in S4 File.

The scripts for these analyses, along with detailed annotations, are available at https://doi.org/10.5281/zenodo.16729059 [27].

### Ethical statements

This study was approved by the Scientific Committee of Universidad de la Costa. However, no humans, animals, or medical records were used as units of analysis.

### Results

Among the 2639 articles published across the four World Bank income groups, HICs accounted for the majority of publications in the medical errors field (83.16%), followed by UMICs with 11.1%, LMICs with 5.2%, and LICs contributing only eleven publications (0.4%). Additionally, HICs produced the most cited publications (36% of all citations) and were the first to publish in this field, with their initial publication appearing in 1995 (Fig 2).

HICs also demonstrated the highest average H-index (98.4). Interestingly, LICs ranked second in H-index (53.2), followed by LMICs (50.7). LICs and LMICs also showed the highest ratios of open access articles (1.75 and 1.24, respectively), while HICs had the lowest open access ratio (0.52) (Table 2).

Regarding document types, the most common across all income groups were articles, followed by reviews. LICs reported the highest proportion of articles (100%), followed by UMICs (85.4%). In contrast, reviews were most frequent in HICs (19%) and LMICs (10.8%). Most publications from HICs were published in Q1 (50.4%) and Q2 (24.4%) journals. Conversely, UMICs and LMICs had the highest proportion of publications in Q2 (28%) and Q3 (36%) journals,

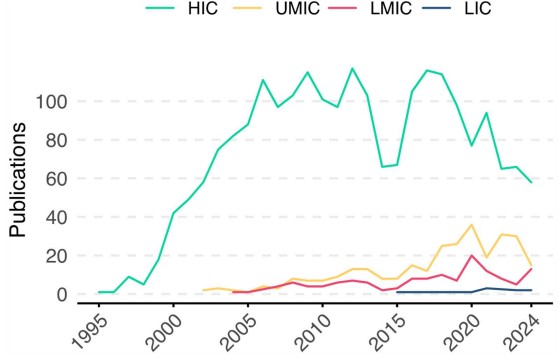

**Fig 2. Temporal trends in medical error publications by country income classification (1995-2024).** Annual scientific publication output on medical errors stratified by World Bank income classification (HIC: high-income countries, UMC: upper-middle-income countries, LMIC: lower-middle-income countries, LIC: low-income countries). The y-axis represents the absolute number of publications per year, while the x-axis shows the publication year.

**Table 2. General bibliometric characteristics by income group (N = 2639).**

| | High-income | Upper-middle-income | Low- and-middle-income | Low-income |
|---|---|---|---|---|
| **Publications (%)** | 2198 (83.16) | 295 (11.16) | 139 (5.26) | 11 (0.42) |
| **Average H-index (SD)** | 98.4 (106.8) | 49.1 (105.5) | 50.7 (99.1) | 53.2 (86.5) |
| **Open Access/ No Open Access** | 758/ 1440 | 142/ 153 | 77/ 62 | 7/ 4 |
| **Document type (%)** | | | | |
| Article | 1449 (65.9) | 252 (85.4) | 111 (79.9) | 11 (100) |
| Editorial | 91 (4.1) | 4 (1.4) | 3 (2.2) | 0 |
| Letter | 93 (4.2) | 12 (4.1) | 6 (4.3) | 0 |
| Note | 78 (3.5) | 2 (0.7) | 3 (2.2) | 0 |
| Review | 417 (19) | 25 (8.5) | 15 (10.8) | 0 |
| Short Survey | 70 (3.2) | 0 | 1 (0.7) | 0 |
| **Journal quartile (%) (n = 2332)** | | | | |
| Q1 | 981 (50.4) | 64 (24.5) | 23 (20.2) | 3 (30) |
| Q2 | 476 (24.4) | 73 (28) | 29 (25.4) | 4 (40) |
| Q3 | 287 (14.7) | 63 (24.1) | 41 (36) | 3 (30) |
| Q4 | 203 (10.4) | 61 (23.4) | 21 (18.4) | 0 |

SD: Standard deviation.

respectively. The highest proportion of Q4 journal publications was observed in UMICs (23.4%) and LMICs (18.4%), whereas LICs reported no publications in Q4 journals.

## Mortality indicators

For this domain, six indicators were examined as dependent variables to measure the potential impact of medical errors research on them. In general, all of these indicators showed a decreasing pattern over the years, except for the crude death rate (per 1,000 people), which showed a small increase in recent years in HICs and UMICs (Figure S1 in S5 File). Interestingly, when modeling medical-errors research with this same indicator, no significant effect was observed across any of the income levels.

However, for mortality rates in neonatal (β = −0.39 to −0.02), under-5 (β = −1.5 to −0.04), infant (β = −0.9 to −0.03), and adult male (β = −1.2 in UMICs and −0.4 in HICs) and female (β = −3.4 to −0.2) populations, our models suggested a potentially significant protective effect for each new article published in countries ranging from LMICs to HICs. Notably, although HICs produced the majority of publications, the models revealed a relatively stronger protective effect in lower income levels (Fig 3). Furthermore, no significant results were observed in LICs, possibly due to the low number of publications from these regions.

## Health system indicators

In this domain, we first used the number of physicians (per 1,000 people) as an independent variable in the models to estimate a proxy of medical errors scientific publication production specifically by medical professionals across income groups. Interestingly, this proxy showed that each additional physician in HICs, an indicator that has proportionally increased by 59.1% in recent years (Figure S2 in S5 File), generated approximately a 154% increase in the expected number of new publications related to medical errors (incidence rate ratio = 2.54, 95% CI: 1.33–4.85, p < 0.05), while in UMICs, with a proportionally lower increase of 12.4% in physician density compared to HICs, each new physician was

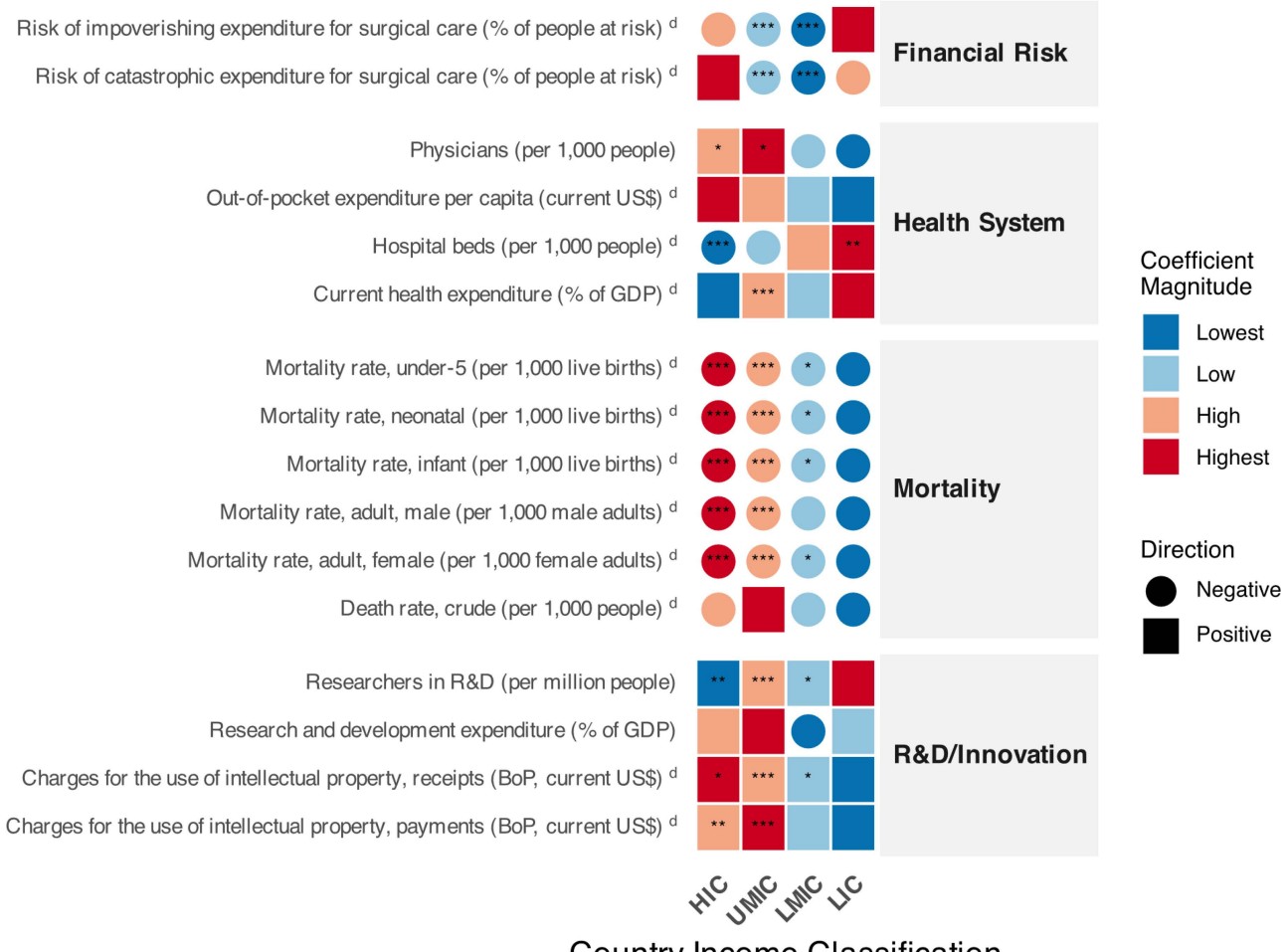

**Fig 3. Relationship between medical errors research output and health indicators across country income classifications.** The matrix displays associations between medical error publication output and 16 indicators across four country income groups (HIC: high-income countries, UMCI: upper-middle-income countries, LMIC: lower-middle-income countries, LIC: low-income countries). Squares represent positive associations and circles represent negative associations. Color intensity indicates the magnitude of effect within each indicator, ranging from lowest (blue) to highest (red). Statistical significance is denoted by asterisks (* $p < 0.05$, ** $p < 0.01$, *** $p < 0.001$). Indicators marked with superscript "d" were analyzed as dependent variables with publications as the predictor; unmarked indicators served as predictors of publication output.

associated with a potential increase of 425% (incidence rate ratio = 5.25, 95% CI: 1.71–16.0, $p < 0.05$) (Fig 3). Notably, this physician-driven research relationship did not show significant results for lower income levels where, in addition, this indicator has shown a proportional reduction.

We also assessed the potential impact of this topic of research on the out-of-pocket expenditure per capita (current US$), an indicator that had an increase of 197.5% in recent years (Figure S2 in S5 File) in HICs. However, we did not find potential associations based on adjusted p-values. Additionally, we did not observe any significant association with the current health expenditure as percentage of Gross Domestic Product (GDP), except in UMICs, where this indicator has shown a decrease of 1.7% in recent years; the model estimated a potential increase of 0.4% in this indicator for each additional published article (OR = 1.004, 95% CI: 1.002–1.006, $p < 0.001$) (Fig 3).

Lastly, in this domain, we evaluated the potential association with hospital beds per 1,000 people as dependent variable, an indicator that has shown a decrease in HICs and UMICs, and an increase in lower income levels. We estimated

that each new article decreased the number of hospital beds by 0.019 per 1,000 people in HICs (β = −0.019, 95% CI: −0.02 to −0.01, p < 0.001) and increased them by 0.16 in LICs (β = 0.16, 95% CI: 0.12–0.20, p < 0.01), with no other significant associations (Fig 3).

### R&D and innovation indicators

Regarding these indicators, we aimed to first assess how large were the impact of R&D expenditure (as percentage of GDP) as a driver for medical error research output. This indicator had an important decrease in HICs coupled with an increase in LICs. However, we found that this indicator did not either increase or decrease significantly the research output.

Additionally, just as we did with physician's density indicator, we estimated a proxy of scientific production using the number of researchers in R&D per million people, finding that each new researcher on this R&D area was associated with a potential increase of 0.3% in the expected number of published papers in UMICs (incidence rate ratio = 1.003, 95% CI: 1.002–1.004, p < 0.001) (Fig 3), being this the highest effect size among the income levels, in which the proxy for LICs was not significant. Interestingly, this indicator has shown a relatively constant increase along years (Figure S3 in S5 File).

Lastly in this domain, we modelled the charges for the use of intellectual property either as payments or receipts (Balance of Payment [BoP], current US$). These indicators have shown an increase along years, more markedly in HICs (Figure S3 in S5 File). We estimated a potential increase of $35.2 (95% CI: 10.3–59.9) million per each medical error article in the intellectual property receipts in HICs (p < 0.05), of $11.7 (95% CI: 7.3–16.0) million in UMICs (p < 0.001) and of $3.26 (95% CI: 0.5–5.9) million in LMICs (p < 0.05), with no association in LICs (Fig 3), with charges for the use of intellectual property as receipts. Similarly, the potential increase per each new article ranged from $42.5 (95% CI: 25.2–59.7) million (p < 0.001) in UMICs to $33.7 (95% CI: 12.1–55.2) million (p < 0.01) in HICs (Fig 3), with no other significant result, with charges for the use of intellectual property as payments.

### Financial surgical risk

Within this domain, we explored the potential impact of medical errors research output on, first, the risk of impoverishing expenditure for surgical care (% of people at risk), an indicator that has shown a relative decrease over the years in every income level (Figure S4 in S5 File).

However, we found significant results for this indicator only in middle income levels, with each new article being associated with a potential reduction of 3.3% (OR: 0.97, 95% CI: 0.96–0.98, p < 0.001) in UMICs to 4.7% (OR: 0.96, 95% CI: 0.94–0.98, p < 0.001) in LMICs, in the odds of impoverishing expenditure for surgical care (Fig 3). A similar pattern was observed regarding the risk of catastrophic expenditure for surgical care (% of people at risk), with each article being associated with a potential reduction of 4.1% (OR: 0.96, 95% CI: 0.95–0.97, p < 0.001) in UMICs and 4.8% (OR: 0.95, 95% CI: 0.92–0.97, p < 0.001) in LMICs, in the odds of catastrophic expenditure for surgical care (Fig 3).

Hierarchical mixed-effects models examining medical errors publications and mortality indicators revealed associations with under-5 mortality (β = 0.0649, 95% CI: 0.0083–0.1215, adjusted p = 0.127) and infant mortality (β = 0.0364, 95% CI: 0.0032–0.0695, adjusted p = 0.127), both non-significant after multiple comparison adjustment (Fig 4).

No significant associations were found for neonatal mortality (β = 0.0104, 95% CI: −0.0014–0.0221, adjusted p = 0.165), adult female mortality (β = 0.0874, 95% CI: −0.0436–0.2185, adjusted p = 0.281), adult male mortality (β = −0.0149, 95% CI: −0.1270–0.0973, adjusted p = 0.793), or crude death rate (β = −0.0018, 95% CI: −0.0081–0.0045, adjusted p = 0.626). Large random effect variances (126.8–6997.4) indicated substantial baseline differences between income groups (Fig 4).

### Hierarchical analysis of health system indicators

Health system indicators showed a significant negative association between publication output and hospital beds (β = −0.0141, 95% CI: −0.0181 to −0.0102, adjusted p < 0.001), indicating countries with higher medical error research tend to have fewer hospital beds. No significant associations were found for current health expenditure as percentage of GDP

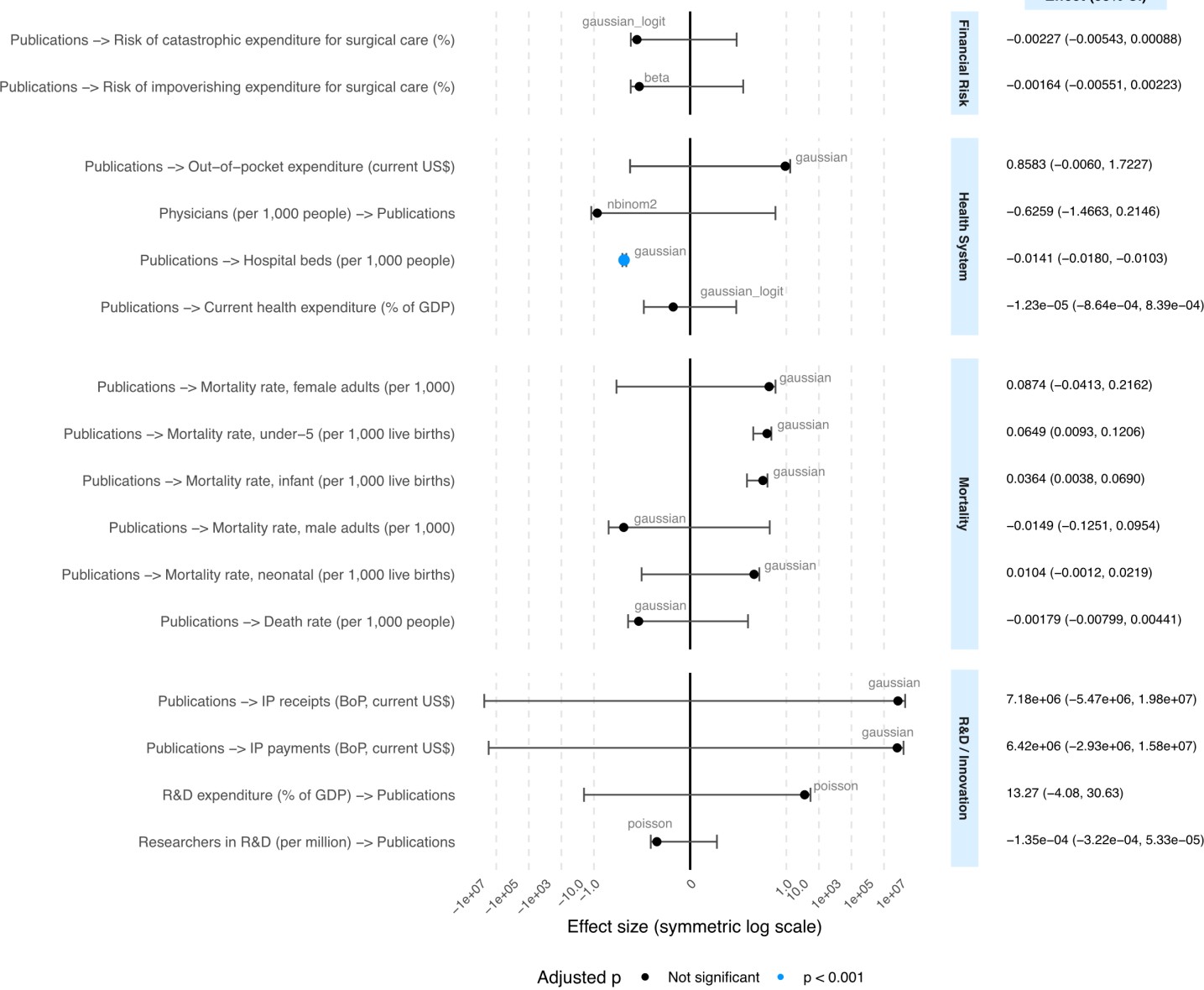

**Fig 4. Hierarchical mixed-effects models of associations between medical error publications and health indicators.** Forest plot displaying effect estimates from hierarchical mixed-effects models examining bidirectional relationships between medical error publication output and health indicators across 195 countries. Points represent regression coefficients with horizontal lines indicating 95% confidence intervals on a symmetric logarithmic scale. Arrow direction (→) indicates the predictor-outcome relationship being tested. Blue points indicate statistical significance at p < 0.001 after Benjamini-Hochberg correction for multiple comparisons; black points represent non-significant associations. Model distribution family is indicated next to each estimate (gaussian, gaussian_logit, poisson, nbinom2, beta). All models included country income level (HIC: high-income countries, UMIC: upper-middle-income countries, LMIC: lower-middle-income countries, LIC: low-income countries) as a random effect to account for clustering and publication year as a covariate. Numerical values of effect estimates and confidence intervals are presented in the right panel.

(β = 1, 95% CI: 0.9–1.0, adjusted p = 0.977) or out-of-pocket expenditure per capita (β = 0.858, 95% CI: −0.022–1.739, adjusted p = 0.134) (Fig 4).

When examining physicians as a predictor of publications, our negative binomial model found no significant relationship (incidence rate ratio = 0.53, 95% CI: 0.23–1.23, adjusted p = 0.144), contrasting with our earlier regression findings by income group (Fig 4).

### Hierarchical analysis of R&D and innovation indicators

Regarding R&D indicators, no significant associations was found between publication output and R&D expenditure as percentage of GDP (incidence rate ratio = 581860, 95% CI: $0.01–2 \times 10^{13}$, adjusted p = 0.134) or researchers in R&D per million people (incidence rate ratio = 0.99, 95% CI: 0.99–1.00, adjusted p = 0.160) (Fig 4).

Similarly, models examining intellectual property charges revealed positive but non-significant associations for both intellectual property receipts (β = $7.18 \times 10^6$, 95% CI: $−5.68 \times 10^6$-$2.00 \times 10^7$, adjusted p = 0.359) and intellectual property payments (β = $6.42 \times 10^6$, 95% CI: $−3.08 \times 10^7$–$1.59 \times 10^7$, adjusted p = 0.281), suggesting potential but statistically non-robust economic value when controlling for income group differences (Fig 4).

### Hierarchical analysis of financial surgical risk

No significant associations were found between medical error publications and risk of catastrophic expenditure (OR: 0.99, 95% CI: 0.99–1.00, adjusted p = 0.163) or impoverishing expenditure for surgical care (OR: 0.99, 95% CI: 0.99–1.00, adjusted p = 0.488) when accounting for income group clustering.

### Discussion

This study provides empirical evidence on the relationship between scientific research on medical errors and a broad set of health, mortality, financial, and research indicators across countries with different income levels. By integrating bibliometric data with global population health- and structural-level indicators over time, the analysis revealed three key findings: 1) research on medical errors has increased significantly since the early 2000s, especially in HICs; 2) only a few population-and structural-level indicators showed statistically significant associations with research activity, particularly in HICs and UMICs; and 3) the overall influence of this body of research on measurable national outcomes remains limited and inconsistent.

The analysis of mortality indicators revealed suggestive patterns linking medical error research with improvements in population health [28], particularly in countries with greater research output. While most indicators showed a declining trend over time, only some demonstrated potential associations with publication volume. Regression models suggested that each additional article on medical errors may be associated with lower rates of neonatal, infant, under-5, and adult mortality, especially among females, across LMICs to HICs. Paradoxically, despite most research originating from HICs, the strongest protective effects were observed in countries with fewer resources, where even small system improvements may yield greater marginal gains.

However, these findings were not robust across modelling strategies. In the hierarchical models, no associations remained statistically significant after adjusting for multiple comparisons, although infant and under-5 mortality approached significance (adjusted p = 0.127). The lack of effect in LICs likely reflects the low volume of scientific production on medical errors in these regions.

The weaker or inconsistent associations observed in LMICs likely reflect structural and epistemic barriers rather than a true absence of effect. These include underreporting of medical errors, limited patient safety surveillance systems, lower regulatory and institutional capacity to translate research into practice, and publication bias that disproportionately favours studies from high-income settings. In addition, the smaller volume and visibility of locally produced research in LMICs may further attenuate detectable associations between scientific output and national indicators.

Altogether, these results suggest that while medical error research may contribute to reducing avoidable mortality, its measurable impact at the national level is uneven and influenced by structural differences in research capacity, policy implementation, and baseline health system performance [29,30].

The influence of medical error research on national outcomes is likely to operate through indirect and cumulative mechanisms rather than through immediate causal effects. These include the diffusion of scientific evidence into clinical guidelines, patient safety regulations, accreditation standards, and quality improvement programs, as well as the gradual development of a culture of safety within health systems. Through these pathways, research on medical errors can shape professional behaviour, organizational practices, and policy priorities, which in turn may contribute to changes in mortality, financial protection, and system efficiency over time.

The association between research on medical errors and health system indicators revealed contrasting patterns across income levels. In HICs and UMICs, where the number of physicians per 1,000 people has grown notably in recent years, a stronger physician presence appeared to correlate with increased research activity. Specifically, each additional physician in HICs and UMICs was associated with approximately 154% and 425% increase in publications on medical errors, respectively (both $p < 0.05$).

However, this physician-driven relationship was absent in LICs, where physician density has either stagnated or declined. These findings suggest that the growth of the medical workforce may contribute to knowledge production in patient safety [31], though unevenly distributed across the global income spectrum.

Beyond workforce metrics, other health system indicators revealed mixed results. No significant associations were observed between research output and out-of-pocket spending or overall health expenditure as a share of GDP, with the exception of UMICs, where a small positive effect on health expenditure was identified ($p < 0.001$). Notably, a consistent inverse relationship was found between publication volume and hospital bed density, confirmed in both income-stratified and hierarchical models, indicating that countries with greater medical error research tend to report fewer hospital beds per 1,000 people (adjusted $p < 0.001$).

While this may reflect shifts toward outpatient care or improved care efficiency [32], it also underscores the need to interpret such structural indicators within context, as they may not directly capture safety outcomes but rather broader system adaptations over time [33].

The relationship between research on medical errors and national capacities for innovation and research revealed selective associations. Although R&D expenditure as a percentage of GDP did not show a significant effect on research output across income groups, the number of researchers in R&D per million people emerged as a potential proxy of productivity, particularly in UMICs, where each additional researcher was associated with an increase of 0.3% in the expected number of papers ($p < 0.001$).

While modest in magnitude, this finding underscores the relevance of human capital investment in bolstering research efforts on patient safety [34]. The absence of association in LICs, where the number of R&D professionals is markedly lower, highlights a persistent gap in capacity that may limit the emergence of locally driven research agendas [35,36].

When examining economic indicators related to intellectual property, the models suggested a possible relationship between publication activity and national revenue from intellectual property receipts, especially in HICs. For example, each new article on medical errors was associated with estimated increases of $35.2 million in HICs, $11.7 million in UMICs, and $3.26 million in LMICs in intellectual property receipts (all $p < 0.05$), although these associations did not remain significant in hierarchical models.

Similar upward trends were noted for intellectual property payments. These patterns may reflect an indirect signal of innovation ecosystems where research on safety is part of broader knowledge generation, regulation, and commercialization processes [37]. However, the lack of statistical robustness after adjusting for income group effects suggests that while medical error research may align with innovation dynamics, it is unlikely to be a primary driver of economic outputs from intellectual property at the national level [38].

Among the most compelling findings of this study were those related to financial protection in the context of surgical care. In UMICs and LMICs, each additional publication on medical errors was associated with a notable reduction in the odds of catastrophic and impoverishing expenditures due to surgery. Specifically, each new article corresponded to an estimated 4.1% to 4.8% reduction in the odds of impoverishing, and 3.3% to 4.7% reduction in the odds of catastrophic expenditure (all p < 0.001).

These findings, while observational, suggest that increased research attention to medical errors may align with improved patient safety practices that ultimately reduce the financial consequences of adverse surgical events [39], particularly in settings where safety lapses may more frequently lead to economic hardship.

However, when adjusting for country-level heterogeneity in the hierarchical models, these associations did not remain statistically significant. This discrepancy reflects the broader pattern observed across other domains: promising associations at the income-stratified level that weaken once structural differences between countries are considered [9]. Even so, the consistent effect direction and the magnitude of change observed in middle-income countries point to an area of potential leverage.

These regions may represent a sweet spot where health systems are developed enough to apply evidence from medical error research, yet still vulnerable enough for improvements in safety to result in measurable financial protection [40]. As such, the findings highlight a critical opportunity to integrate patient safety strategies into broader efforts to enhance financial risk protection in global surgery agendas.

These results suggest that the global agenda on medical errors may be progressing unevenly, with HICs benefiting from mature feedback loops between research, policy, and practice, while low-income settings remain disconnected from these cycles, both in production and implementation [41].

The observed associations could also reflect reverse causality: countries with better-resourced health systems and research institutions may naturally produce more research, including on safety. Alternatively, shared enabling factors (e.g., academic networks, regulatory frameworks, or international collaboration) may simultaneously support both research growth and system improvements [42].

From a clinical and policy perspective, these findings highlight the need to rethink how research in patient safety is positioned within national and global agendas. The results underscore that the mere existence of publications may not be sufficient to drive change, especially in countries lacking robust knowledge translation mechanisms or institutional readiness [33,43]. More than ever, efforts to reduce medical errors must be accompanied by structural investments, political commitment, and policy frameworks that support evidence-informed reform.

Future research should explore mechanisms of translation in greater detail, including how findings from safety science are incorporated into guidelines, regulation, and workforce training [44,45]. Mixed-methods approaches and longitudinal policy tracking could offer insights into the pathways that enable or hinder the real-world impact of research. Addressing the equity gap will also require building capacity for research production, dissemination, and use in LMICs and LICs, potentially through collaborative networks and funding mechanisms [46].

Importantly, the associations observed in this study should not be interpreted as evidence of direct causality. Given the ecological and observational design, these relationships may reflect shared structural determinants, reverse causality, or unmeasured confounding rather than causal effects of medical error research on national outcomes. The findings therefore describe patterns of alignment between research activity and health system indicators rather than proof of causal impact.

From a global health perspective, these findings may help inform future patient safety metrics and WHO policy initiatives by illustrating the importance of integrating research production with system-level indicators. Linking scientific output on medical errors with measures of mortality, financial protection, and health system capacity could support more comprehensive monitoring frameworks for patient safety, aligned with universal health coverage and the Global Patient Safety Action Plan [4].

## Limitations

Some limitations should be acknowledged. First, the ecological and observational nature of the study precludes causal inference; associations observed may be influenced by unmeasured confounders such as health policy reforms or institutional practices not captured in national indicators.

Second, the use of scientific publication counts as a proxy for research activity does not account for quality, implementation, or translation into practice. Third, LICs were underrepresented in research output, limiting statistical power and potentially underestimating associations in those contexts. Finally, despite efforts to adjust for heterogeneity, residual confounding related to structural and socio-political factors cannot be ruled out.

Although indicators with high missingness were excluded, remaining incomplete data may have influenced some estimates. Bibliometric databases such as Scopus, Web of Science, and PubMed underrepresent non-English and locally published research, particularly from LICs and LMICs, introducing potential selection bias. Also, reverse causality is possible, as countries with stronger health systems and research infrastructure may both generate more medical error research and achieve better health and financial outcomes.

However, a major strength of this study lies in its novel and integrative design, which combines bibliometric data with a broad set of population-level indicators, analysed longitudinally and stratified by country income level. The use of both stratified and hierarchical modelling approaches strengthens the internal robustness of findings and allows for more nuanced interpretation across diverse contexts.

Moreover, the study addresses a clear gap in the literature, offering empirical insights into the potential population- and health-level influence of medical error research.

## Conclusions

This study provides empirical evidence suggesting that scientific research on medical errors may contribute to measurable improvements in population health, health system structure, and financial protection, particularly in countries with emerging research capacity. Across multiple domains, associations were observed in middle-income settings, where each additional publication was linked to reductions in mortality rates and surgical impoverishment risk, as well as modest but relevant shifts in health system and innovation-related indicators.

These findings confirm that the influence of medical error research can extend beyond institutional or clinical environments, reaching broader population-level outcomes. More importantly, they underscore the inequity in research visibility and impact across income levels. By bridging bibliometric analysis with public health indicators, this study moves the field forward conceptually and methodologically, responding to a longstanding gap in understanding how patient safety research translates into real-world benefit. It reinforces the need for policies and investments that not only support research productivity, but also ensure that such knowledge generates value for health systems and societies at large.

From a policy and global health perspective, these findings suggest that investments in patient safety research should be accompanied by mechanisms for knowledge translation, regulatory uptake, and health system strengthening, particularly in middle-income and low-income settings. Targeted funding, international research partnerships, and integration of safety science into national health strategies may help convert scientific production into measurable improvements in population health and financial protection.

## Supporting information

**S1 File. Full search strategies.**
(DOCX)

**S2 File. Bibliometric data and aggregated indicators by income group and year.**
(XLSX)

**S3 File. Full results of the regression models.**
(XLSX)

**S4 File. Full results of the hierarchical models.**
(XLSX)

**S5 File. Contains supporting figures 1–4.**
(DOCX)

## Author contributions

**Conceptualization:** Mabel Adelvia Sarquis Rivera, David A. Hernandez-Paez, Johana Galván-Barrios, Ernesto Barceló-Martinez, Alexis Rafael Narvaez-Rojas, Ivan David Lozada-Martinez.

**Formal analysis:** David A. Hernandez-Paez, Johana Galván-Barrios.

**Investigation:** Mabel Adelvia Sarquis Rivera, David A. Hernandez-Paez, Johana Galván-Barrios, Alexis Rafael Narvaez-Rojas, Ivan David Lozada-Martinez.

**Methodology:** David A. Hernandez-Paez, Ivan David Lozada-Martinez.

**Supervision:** Ernesto Barceló-Martinez, Alexis Rafael Narvaez-Rojas.

**Writing – original draft:** Mabel Adelvia Sarquis Rivera, David A. Hernandez-Paez, Johana Galván-Barrios, Ernesto Barceló-Martinez, Alexis Rafael Narvaez-Rojas, Ivan David Lozada-Martinez.

**Writing – review & editing:** Mabel Adelvia Sarquis Rivera, David A. Hernandez-Paez, Johana Galván-Barrios, Ernesto Barceló-Martinez, Alexis Rafael Narvaez-Rojas, Ivan David Lozada-Martinez.

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
