## [Decision Letter · Decision Letter 0]

8 Jan 2026

Dear Dr. Narvaez-Rojas,

Thank you for submitting your manuscript to PLOS ONE. After careful consideration, we feel that it has merit but does not fully meet PLOS ONE’s publication criteria as it currently stands. Therefore, we invite you to submit a revised version of the manuscript that addresses the points raised during the review process.

We look forward to receiving your revised manuscript.

Kind regards,

Mohamad K. Abou Chaar, M.D.

Academic Editor

PLOS One

Journal Requirements:

2. Please remove your figures from within your manuscript file, leaving only the individual TIFF/EPS image files, uploaded separately. These will be automatically included in the reviewers’ PDF.

Additional Editor Comments:

Rivera et al. have submitted a manuscript titled “Exploring the Potential Impact of Medical Errors Research on Population Health, Health System, and Research and Development Indicators”, in which they examine whether growth in the scientific literature on medical errors is associated with changes in macro-level health, health-system, and innovation indicators over time.

I have the following questions:

- Your eligibility criteria require a “clearly defined objective” related to medical errors and also require full-text availability, but it is not clarified how reviewers operationalized “clearly defined objective,” nor why full text was necessary for a primarily bibliometric/metadata-based dataset.

- You exclude indicators with >25% missingness and then use a left join preserving all bibliometric records. The manuscript does not clarify whether remaining missingness was handled by complete-case analysis, year-wise deletion, imputation, or variable-specific denominators across models.

Reviewers' comments:

Reviewer's Responses to Questions

**Comments to the Author**

1. Is the manuscript technically sound, and do the data support the conclusions?

Reviewer #1: Yes

Reviewer #2: Yes

2. Has the statistical analysis been performed appropriately and rigorously?

Reviewer #1: Yes

Reviewer #2: No

3. Have the authors made all data underlying the findings in their manuscript fully available?

Reviewer #1: Yes

Reviewer #2: Yes

4. Is the manuscript presented in an intelligible fashion and written in standard English?

Reviewer #1: Yes

Reviewer #2: Yes

Reviewer #1: The introduction provides a comprehensive overview of medical errors and their relevance, but the rationale for examining bibliometric research impact on national indicators could be more clearly articulated.

The introduction would benefit from a more explicit articulation of the knowledge gap: what specific aspect of prior evidence is missing? (e.g., temporal analysis, cross-country comparison, link to innovation indicators).

The term “medical errors research” is used frequently but not formally defined. Does it include diagnostic, medication, surgical, and system-level errors? Please clarify.

The discussion of disparities between HICs and LMICs is strong. However, could the authors elaborate on why low- and middle-income countries might exhibit weaker associations — e.g., due to underreporting, lack of safety policies, or publication bias?

The 1995–2024 period is impressive in scope. How did the authors ensure consistency in bibliometric data coverage and indicator definitions across such a long time frame?

The rationale for choosing the 16 final indicators (after excluding six with >25% missing data) should be better justified. Why these specific indicators, and do they adequately represent the four thematic domains?

The statistical approach is complex and well described, but the rationale for using two directions of regression (publications as dependent vs. independent variable) could be better justified conceptually.

The extensive use of abbreviations makes the Results section difficult to read and follow for the reader.

The discussion offers useful reflections but sometimes repeats descriptive findings rather than interpreting mechanisms. Could the authors elaborate on how medical error research might indirectly influence outcomes (e.g., via policy diffusion, quality improvement culture)?

The discussion section would benefit from a clearer distinction between correlation and causation.

The conclusion could be strengthened by proposing actionable implications.

Missing data, selection bias in bibliometric databases (e.g., underrepresentation of non-English literature), and potential reverse causality should be explicitly acknowledged.

Could the authors provide a brief reflection on how this work might inform future global patient safety metrics or WHO policy initiatives?

Reviewer #2: Figure 1: clarify how the sample size decreased from 11,798 screened to 2,639, and specify the reasons for exclusion.

Line 320: A variance-to-mean ratio calculated from the outcome alone (without accounting for predictors) may appear to indicate equidispersion even when the fitted model is overdispersed, and vice versa. Overdispersion is usually evaluated in a model setting.

Line 323: clarify “binomial regression.” Was a logistic regression used?

Line 324/374: specify the transformation used to map proportions into the open interval (0, 1)

Line 331: Fitting separate models within strata can be unstable, particularly with small sample sizes, and may increase the risk of separation and small-sample bias.

Line 333/355: The two primary model sets needs to be justified. Explain why publication counts were used both as an outcome and as a independent variable.

Table 1: The term “MVs” in the regression model is confusing. State that they were excluded rather than listing “MVs” in the model.

**Do you want your identity to be public for this peer review?** For information about this choice, including consent withdrawal, please see our Privacy Policy

Reviewer #1: No

Reviewer #2: No

---

## [Author Response · Author response to Decision Letter 1]

12 Jan 2026

Answer to reviewers

PONE-D-25-42260

Exploring the Potential Impact of Medical Errors Research on Population Health, Health System, and Research and Development Indicators

We are grateful with the Editor and Reviewers for the expert reading of the manuscript. We have found the comments appropriate and helpful. They have permitted us to improve the manuscript. Appropriate changes were made and highlighted in the revised manuscript according to suggestions. Following are the responses to the comments.

Editor comments

r/ Corrected.

2. Please remove your figures from within your manuscript file, leaving only the individual TIFF/EPS image files, uploaded separately. These will be automatically included in the reviewers’ PDF.

r/ Corrected.

r/ Understood, editor.

4. Your eligibility criteria require a “clearly defined objective” related to medical errors and also require full-text availability, but it is not clarified how reviewers operationalized “clearly defined objective,” nor why full text was necessary for a primarily bibliometric/metadata-based dataset.

r/ We thank the Editor for this important request for clarification. We agree that the operationalization of “clearly defined objective” and the rationale for requiring full-text availability should be made explicit.

In our study, a “clearly defined objective related to medical errors” was operationalized as a publication in which the primary stated aim (in the title, abstract, or objectives section) was to analyze, measure, classify, interpret, or evaluate medical errors, patient safety events, or error-related processes in health care. Articles in which medical errors were only mentioned tangentially (e.g., in background sections, examples, or secondary analyses) were excluded.

Full-text availability was required not because we extracted full-text data for bibliometric analyses, but to allow reviewers to verify that the publication genuinely met this objective criterion and was not a false positive generated by keyword matching alone. This step ensured the thematic validity and internal consistency of the bibliometric dataset.

We have now clarified both points in the Methods section.

5. You exclude indicators with >25% missingness and then use a left join preserving all bibliometric records. The manuscript does not clarify whether remaining missingness was handled by complete-case analysis, year-wise deletion, imputation, or variable-specific denominators across models.

r/ We appreciate this request for clarification regarding the handling of missing data after dataset integration. After excluding indicators with more than 25% missingness, the remaining missing values were handled using an available-case (variable-specific denominator) approach.

Specifically, after the left join that preserved all bibliometric records, each regression model was fitted using all available observations for the specific indicator, year, and income group involved in that model. No imputation was performed, and no global complete-case deletion was applied across variables, as this would have unnecessarily reduced statistical power and introduced bias.

We have now explicitly stated this in the Data Synthesis and Statistical Analysis sections.

Reviewer #1

The introduction provides a comprehensive overview of medical errors and their relevance, but the rationale for examining bibliometric research impact on national indicators could be more clearly articulated. The introduction would benefit from a more explicit articulation of the knowledge gap: what specific aspect of prior evidence is missing? (e.g., temporal analysis, cross-country comparison, link to innovation indicators).

r/ We agree that the knowledge gap motivating this study should be stated more explicitly. While prior research has extensively examined medical errors at the clinical and institutional level, there is currently no longitudinal, cross-country, and multi-domain evaluation of whether scientific research on medical errors is associated with changes in population health, health system performance, financial protection, or innovation indicators.

We have now clarified this gap in the Introduction by explicitly stating that previous evidence is limited to local or clinical outcomes and does not address the temporal, international, and structural dimensions of research impact that this study examines.

The term “medical errors research” is used frequently but not formally defined. Does it include diagnostic, medication, surgical, and system-level errors? Please clarify.

r/ We appreciate this important request for clarification. In this study, “medical errors research” was defined as scientific publications whose primary objective was to analyze, measure, classify, or evaluate errors occurring in the delivery of health care. This includes diagnostic errors, medication errors, procedural and surgical errors, and system-level or organizational safety failures.

This definition was operationalized through our search strategy, which incorporated controlled vocabulary terms and keywords such as “medical errors,” “diagnostic errors,” and “medication errors,” and was applied across multiple biomedical databases. We have now added an explicit definition of this construct in the Methods section to avoid ambiguity.

The discussion of disparities between HICs and LMICs is strong. However, could the authors elaborate on why low- and middle-income countries might exhibit weaker associations — e.g., due to underreporting, lack of safety policies, or publication bias?

r/ We agree that the weaker or inconsistent associations observed in low- and middle-income countries require further explanation. These patterns are likely driven by several structural mechanisms, including underreporting of medical errors, weaker patient safety surveillance systems, limited policy uptake of research findings, lower research capacity, and publication bias against LMIC-based studies.

We have now expanded the Discussion to explicitly address these factors and to clarify that differences in observed associations reflect not only research volume, but also disparities in knowledge translation and health system readiness.

The 1995–2024 period is impressive in scope. How did the authors ensure consistency in bibliometric data coverage and indicator definitions across such a long time frame?

r/ To ensure comparability across the 1995–2024 period, we relied on standardized, internationally curated data sources for both bibliometric and indicator data. Bibliometric records were retrieved using reproducible search strategies applied across five major databases with stable indexing practices, while all health, financial, and R&D indicators were obtained from the World Bank and the World Health Organization’s Global Observatory on Health Research and Development, which use harmonized definitions and retrospective normalization across years.

In addition, all indicators were extracted in a single retrieval window (July 2024) to ensure internal consistency, and country income classifications were standardized using the 2024 World Bank grouping. These procedures ensure that observed temporal patterns reflect real changes rather than shifts in data definitions or coverage. We have clarified this in the Methods section.

The rationale for choosing the 16 final indicators (after excluding six with >25% missing data) should be better justified. Why these specific indicators, and do they adequately represent the four thematic domains?

r/ The final set of 16 indicators was not selected ad hoc, but derived from a predefined conceptual framework representing four complementary domains through which medical error research could plausibly influence society: mortality (population health), health system capacity, financial risk protection, and research and innovation.

Within each domain, we initially included all relevant indicators available from the World Bank and WHO Global Observatory databases. Indicators with more than 25% missingness were excluded to preserve statistical validity and longitudinal stability, leaving a set of indicators that were both theoretically meaningful and empirically robust.

We have now clarified this rationale in the Methods section.

The statistical approach is complex and well described, but the rationale for using two directions of regression (publications as dependent vs. independent variable) could be better justified conceptually.

r/ The bidirectional modeling strategy was intentional and reflects two distinct but complementary meta-research questions: 1) whether scientific production on medical errors is associated with changes in population-level and structural indicators (research impact); and 2) whether health system and R&D characteristics are associated with the volume of medical error research produced (determinants of research production).

These two directions do not represent reverse causality within a single causal model, but rather two analytically separate processes that together characterize the research-system relationship. We have clarified this conceptual framework in the Methods section.

The extensive use of abbreviations makes the Results section difficult to read and follow for the reader.

r/ We agree that the extensive use of abbreviations may reduce readability in the Results section. We have revised the text to expand abbreviations at first mention and to preferentially use full indicator names in the narrative description, while retaining abbreviations in tables and figures where space is limited.

The discussion offers useful reflections but sometimes repeats descriptive findings rather than interpreting mechanisms. Could the authors elaborate on how medical error research might indirectly influence outcomes (e.g., via policy diffusion, quality improvement culture)?

r/ We agree that the Discussion should more clearly articulate the mechanisms through which medical error research could influence population-level and structural outcomes. We have therefore expanded the Discussion to explicitly describe indirect pathways, including policy diffusion, the incorporation of evidence into clinical guidelines, quality improvement initiatives, regulatory changes, and the development of patient safety culture. These mechanisms help explain how scientific output may translate into system-level effects even in the absence of direct causal attribution.

The discussion section would benefit from a clearer distinction between correlation and causation.

r/ We agree that a clearer distinction between correlation and causation is essential in an ecological and observational study such as this. We have therefore revised the Discussion and Limitations sections to explicitly state that the observed relationships represent statistical associations and do not imply direct causal effects, as they may be influenced by unmeasured confounders, reverse causality, and shared structural determinants.

The conclusion could be strengthened by proposing actionable implications.

r/ We have strengthened the Conclusion by adding brief actionable implications highlighting how policymakers, funding agencies, and global health organizations could use these findings to guide investments in patient safety research, knowledge translation, and capacity building, particularly in middle- and low-income countries.

Missing data, selection bias in bibliometric databases (e.g., underrepresentation of non-English literature), and potential reverse causality should be explicitly acknowledged.

r/ We have expanded the Limitations section to explicitly acknowledge three key sources of bias: incomplete indicator coverage and remaining missing data, selection bias in bibliometric databases (including underrepresentation of non-English and LMIC-based literature), and the possibility of reverse causality whereby stronger health systems and research environments produce more medical error research. These clarifications strengthen the transparency and interpretability of our findings.

Could the authors provide a brief reflection on how this work might inform future global patient safety metrics or WHO policy initiatives?

r/ We have added a brief reflection in the Discussion on how our findings could inform future global patient safety metrics and WHO policy initiatives, particularly by highlighting the importance of linking research production, knowledge translation, and system-level indicators in monitoring progress toward safer care.

Reviewer #2

Figure 1: clarify how the sample size decreased from 11,798 screened to 2,639, and specify the reasons for exclusion.

r/ The large reduction from 11,798 screened records to 2,639 included studies was primarily driven by the title and abstract screening stage, during which records were excluded because their primary objective did not meet our operational definition of medical errors research. Specifically, many articles mentioned errors or safety only tangentially but did not focus on the analysis, measurement, or evaluation of medical errors as a core research objective. We have now clarified this in the Figure 1 legend and Methods section.

Line 320: A variance-to-mean ratio calculated from the outcome alone (without accounting for predictors) may appear to indicate equidispersion even when the fitted model is overdispersed, and vice versa. Overdispersion is usually evaluated in a model setting.

r/ We agree that overdispersion is most appropriately evaluated within a fitted model. While variance-to-mean ratios were used as an initial screening heuristic, final model selection between Poisson and negative binomial regression was based on model-based diagnostics, including residual deviance, dispersion statistics, and Akaike Information Criterion. We have revised the Methods section to clarify this point.

Line 323: clarify “binomial regression.” Was a logistic regression used?

r/ Yes, binomial regression in this study refers to logistic regression with a binomial error distribution and logit link function. We have revised the Methods section to explicitly state this and avoid ambiguity.

Line 324/374: specify the transformation used to map proportions into the open interval (0, 1)

r/ Proportional outcomes were transformed using a standard continuity correction to map values into the open interval (0,1) before fitting beta regression models. Specifically, observed proportions were adjusted using a commonly applied transformation that shifts 0 and 1 values slightly inward based on sample size. We have now specified this in the Methods section.

Line 331: Fitting separate models within strata can be unstable, particularly with small sample sizes, and may increase the risk of separation and small-sample bias.

r/ We agree that fitting separate income-stratified models may be unstable in small samples. For this reason, we complemented stratified analyses with hierarchical mixed-effects models that pool information across income groups while accounting for between-group heterogeneity. This approach mitigates small-sample bias, separation, and overfitting. We have clarified this rationale in the Methods section.

Line 333/355: The two primary model sets needs to be justified. Explain why publication counts were used both as an outcome and as a independent variable.

r/ The two primary model sets were designed to address two distinct but complementary meta-research questions: 1) whether medical error research output is associated with changes in health, financial, and innovation indicators (publications as predictor); and 2) whether health system and R&D characteristics are associated with the volume of medical error research produced (publications as outcome). These represent analytically separate processes rather than inverse specificatio

---

## [Decision Letter · Decision Letter 1]

27 Feb 2026

Exploring the Potential Impact of Medical Errors Research on Population Health, Health System, and Research and Development Indicators

PONE-D-25-42260R1

Dear Dr. Narvaez-Rojas,

We’re pleased to inform you that your manuscript has been judged scientifically suitable for publication and will be formally accepted for publication once it meets all outstanding technical requirements.

Kind regards,

Mohamad K. Abou Chaar, M.D.

Academic Editor

PLOS One

Additional Editor Comments (optional):

Thank you for addressing all the comments.

Reviewers' comments:

Reviewer's Responses to Questions

**Comments to the Author**

Reviewer #2: All comments have been addressed

Reviewer #3: All comments have been addressed

2. Is the manuscript technically sound, and do the data support the conclusions?

Reviewer #2: (No Response)

Reviewer #3: (No Response)

3. Has the statistical analysis been performed appropriately and rigorously?

Reviewer #2: (No Response)

Reviewer #3: (No Response)

4. Have the authors made all data underlying the findings in their manuscript fully available?

Reviewer #2: (No Response)

Reviewer #3: (No Response)

5. Is the manuscript presented in an intelligible fashion and written in standard English?

Reviewer #2: (No Response)

Reviewer #3: (No Response)

Reviewer #2: All comments are addressed.

Reviewer #3: (No Response)

**Do you want your identity to be public for this peer review?** For information about this choice, including consent withdrawal, please see our Privacy Policy

Reviewer #2: No

Reviewer #3: No

---

## [Editor Report · Acceptance letter]

PONE-D-25-42260R1

PLOS One

Dear Dr. Narvaez-Rojas,

I'm pleased to inform you that your manuscript has been deemed suitable for publication in PLOS One. Congratulations! Your manuscript is now being handed over to our production team.

Kind regards,

on behalf of

Dr. Mohamad K. Abou Chaar

Academic Editor

PLOS One